# The Long-Term Immunogenicity of mRNABNT162b Third Vaccine Dose in Solid Organ Transplant Recipients

**DOI:** 10.3390/vaccines12030224

**Published:** 2024-02-22

**Authors:** Maria Antonella Zingaropoli, Mariasilvia Guardiani, Federica Dominelli, Eeva Tortellini, Manuela Garofalo, Francesco Cogliati Dezza, Anastasia Centofanti, Carolina Carillo, Anna Napoli, Federico Venuta, Claudio Maria Mastroianni, Renzo Pretagostini, Miriam Lichtner, Maria Rosa Ciardi, Gianluca Russo

**Affiliations:** 1Department of Public Health and Infectious Diseases, Sapienza University of Rome, 00185 Rome, Italy; mariasilvia.guardiani@uniroma1.it (M.G.); federica.dominelli@uniroma1.it (F.D.); eeva.tortellini@uniroma1.it (E.T.); francesco.cogliatidezza@uniroma1.it (F.C.D.); claudio.mastroianni@uniroma.it (C.M.M.); maria.ciardi@uniroma1.it (M.R.C.); gianluca.russo@uniroma1.it (G.R.); 2General Surgery and Organ Transplantation Unit, Sapienza University of Rome, 00161 Rome, Italy; manuela.garofalo@uniroma1.it (M.G.); renzo.pretagostini@uniroma1.it (R.P.); 3Department of General and Specialistic Surgery “Paride Stefanini”, Sapienza University of Rome, 00161 Rome, Italy; anastasia.centofanti@uniroma1.it (A.C.); c.carillo@policlinicoumberto1.it (C.C.);; 4Department of Molecular Medicine, Sapienza University of Rome, 00161 Rome, Italy; anna.napoli@uniroma1.it; 5Department of Neurosciences, Mental Health, and Sense Organs, NESMOS, Sapienza University of Rome, 00185 Rome, Italy; miriam.lichtner@uniroma1.it

**Keywords:** SOT-Rs, humoral response, T-cell response, BNT162b2 vaccine, anti-spike antibody, flow-cytometry, lung transplant recipients, kidney transplant recipients

## Abstract

We investigated humoral and T-cell response to a SARS-CoV-2 mRNA vaccine in solid organ transplant recipients (SOT-Rs) and healthy donors (HDs) before (T0) and after two (T1) and twelve months (T2) since the third dose administration. SOT-Rs were stratified according to the transplanted organ and to the time elapsed since the transplant. In SOT-Rs, detectable levels of anti-S antibodies were observed in 44%, 81% and 88% at T0, T1 and T2, respectively. Conversely, anti-S antibody levels were detected in 100% of HD at all time points. Lower antibody titers were observed in SOT-Rs compared to HDs, even stratifying by transplanted organs and the time elapsed since transplant. Lower percentages of responding and polyfunctional T-cells were observed in SOT-Rs as well as in each subgroup of SOT-Rs compared to HDs. At both T0 and T1, in SOT-Rs, a predominance of one cytokine production shortly was observed. Conversely, at T2, a dynamic change in the T-cells subset distribution was observed, similar to what was observed in HDs. In SOT-Rs, the third dose increased the rate of seroconversion, although anti-S levels remained lower compared to HDs, and a qualitatively inferior T-cell response to vaccination was observed. Vaccine effectiveness in SOT-Rs is still suboptimal and might be improved by booster doses and prophylactic strategies.

## 1. Introduction

Since the rapid development and use of vaccines against severe acute respiratory syndrome (SARS-CoV-2), several studies have been conducted in immunocompetent and immunocompromised subjects, demonstrating that vaccination is a safe and highly effective strategy against severe outcomes of Coronavirus disease 2019 (COVID-19) [1,2,3]. In this context, mRNA-based vaccines have become the mainstay for the pandemic response, due to their ability to induce robust and protective humoral and cellular responses against SARS-CoV-2 [4,5]. However, in immunocompromised subjects, low rates of anti-Spike (anti-S) antibody levels and moderate vaccine effectiveness have been observed [3,6,7,8].

The immunosuppressive treatments chronically taken by solid organ transplant recipients (SOT-Rs) are considered a risk factor for severe COVID-19 outcomes [3,9,10,11]. As reported for other vaccinations, such as influenza and human papillomavirus (HPV) [12,13,14], SOT-Rs show greatly reduced humoral and impaired T-cell responses after the SARS-CoV-2 mRNA-based vaccine [6,10,15,16]. In general, the lower response to vaccines observed among SOT-Rs leads to the need to improve vaccine immunogenicity in this population group [10,14,17].

In kidney transplant recipients (KT-Rs), it has been shown that low antibody levels after vaccination might correlate to an increased risk for breakthrough infections and hospitalization [18]. Accordingly, SARS-CoV-2 breakthrough infections with severe disease outcomes in fully vaccinated SOT-Rs have been reported [19,20].

During the COVID-19 pandemic, the emergence of SARS-CoV-2 variants from Alpha to Omicron raised concerns about the efficacy of the mRNA-based vaccines [21,22] because of Spike mutations that could facilitate vaccine-driven immunity escape [23]. Although an adequate immune response to the SARS-CoV-2 vaccine has been reported in SOT-Rs following the second and third doses, the duration of immunity conferred by the vaccine and whether the immunity elicited is able to protect individuals from the Omicron variant remain unclear [18].

A crucial issue is to identify fragile subjects who may benefit from the repeated administration of the SARS-CoV-2 vaccine. To date, it is not entirely clear how the transplanted organ and the time since the transplant itself may affect the immunogenicity of the vaccine after a third dose of BNT162b2 in SOT-Rs. Finally, another critical question to be addressed is whether differences in humoral and T-cell response among SOT-Rs from different transplanted organs depend on the various immunosuppressive strategies or are related to the organ itself. For example, KT-Rs have significantly higher serologic responses than LuT-Rs, but the two populations have not been clearly compared face-to-face in studies that have assessed both cellular and antibody responses [24].

In this framework, the aim of the study was to investigate humoral and cellular immune responses after one year since the third dose of SARS-CoV-2 mRNA-BNT162b2 vaccine in KT-Rs and LuT-Rs, with a special focus on the qualitative characterization of T-cell response against the Omicron variant.

## 2. Materials and Methods

### 2.1. Study Population

SOT-Rs and age- and sex-matched healthy donors (HDs) were enrolled. Specifically, HDs were healthcare workers. According to the Italian national vaccination program [25], at the time of the enrollment both SOT-Rs and HDs received two doses of the SARS-CoV-2 mRNA-BNT162b2 vaccine. For both SOT-Rs and HDs, exclusion criteria were age < 18 years and prior history of SARS-CoV-2 infection.

As reported in Figure 1, blood samples were taken before the third vaccine dose (T0) and then two (T1) and 12 months (T2) following the third vaccine dose. SOT-Rs were evaluated for demographics, comorbidities, basic laboratory findings, months elapsed since transplant, and immunosuppression therapy.

At first, the specific humoral and T-cell response among SOT-Rs was compared to HDs at each time point. Then, SOT-Rs were stratified according to transplanted organs into two subgroups: lung transplant recipients (LuT-Rs) and kidney transplant recipients (KT-Rs), and the differences related to the specific humoral and T-cell response were evaluated. Moreover, SOT-Rs were stratified according to the time elapsed since transplant into less than 2 years and more than 2 years since transplant.

Finally, a further comparison between SOT-Rs and HDs was made at T2 to estimate the specific T-cell response to the Omicron variant as well as to wild-type SARS-CoV-2.

### 2.2. Serological and Specific T-Cell Assessment 

As previously described [7], to rule out previous SARS-CoV-2 infection, in both SOT-Rs and HDs, we assayed specific serum anti-nucleocapsid (N) antibodies to SARS-CoV-2 using the KT-1032 EDI TM Novel Coronavirus COVID-19 IgG Enzyme Linked Immuno-sorbent Assay (ELISA) kit (Epitope Diagnostics, Inc., 7110 Carroll Rd, San Diego, CA, USA). In addition, at all time-points on the collected serum samples, we measured total anti-S SARS-CoV-2 IgG antibodies using the commercial chemiluminescence assay (CLIA) DiaSorin Liaison SARS-CoV-2 TrimericS IgG (DiaSorin TriS IgG; DiaSorin S.p.A, Saluggia, Italy). Anti-SARS-CoV-2 IgG antibody levels were expressed according to the WHO international standard (NIBSC code 20/268) with arbitrary binding unit (BAU/mL) [7].

As previously described [26], peripheral blood mononuclear cells (PBMCs) were isolated, cryopreserved, and stored at −196 °C until used.

T-cell-specific response was assessed using a multiparametric flow cytometry after overnight stimulation, as previously described [8,27]. Moreover, to assess the Omicron variant T-cell response, PepTivator SARS-CoV-2 Prot_S B.1.1.529/BA.5 Mutation Pool (Milteny Biotec, Bergisch Gladbach, Germany) that covers selectively mutated regions was used to supplement PepTivators covering the sequence of the wild type of spike glycoprotein (“S”) in order to detect immune responses towards all three variants: Wuhan (wild type), B.1.1.529/BA.4, and B.1.1.529/BA.5 variant.

As reported in Figure 1, for each participant, an unstimulated and a positive phy-tohemagglutinin (PHA) control was included too. Brefeldin A was added to the culture after 1 h of incubation. After overnight stimulation, PBMC was incubated with Fixable Viability Dye and stained, according to the manufacturer’s instruction, with a mix of fluorochrome-conjugated antibodies. The following fluorochrome-conjugated antibodies were used: anti-CD45, anti-CD4, anti-CD8, anti-IFNg, anti-TNFa and anti-IL2 (Bio-Legend, San Diego, CA, USA). Finally, PBMC was fixed in Phosphate-Buffered Saline (PBS) containing 0.5% formaldehyde (Sigma-Aldrich, St. Louis, MO, USA). Then, all the stained samples were acquired using MACSQuant (Miltenyi Biotec, Bergisch Gladbach, Germany) and analyzed using FlowJo™ v10.8.1 software. For each stained sample, the cytokine background of the unstimulated condition was subtracted from the stimulated ones. Finally, using the Boolean gate, all possible combinations of intracellular expression of IFNγ, IL2, and TNFα in cytokine-producing T-cells were evaluated. We defined “responding T-cells” as those cells that produced any of IFNγ, IL2, and TNFα and “polyfunctional T-cells” as those simultaneously producing all three cytokines. Display and analysis of the different cytokine combinations were performed with SPICE v6.1.

### 2.3. Statistical Analysis

Statistical analyses were performed using GraphPad Prism v.9. Two-tailed *p* ≤ 0.05 was considered statistically significant. All data are reported as median and interquartile range (IQR). The differences between SOT-Rs and HDs as well as between KT-Rs and LuT-Rs were evaluated by a two-tailed Mann–Whitney test for quantitative variables. Conversely, differences among SOT-R subgroups and HDs were investigated by a non-parametric Kruskal–Wallis test with Dunn’s multiple comparison post-test for quantitative variables. Longitudinal evaluation was performed using the nonparametric Wilcoxon test.

## 3. Results

### 3.1. Study Population and Sample Collection

From September 2022 to September 2023, blood samples were collected at T0 and T1 for 32 SOT-Rs and 12 HDs while, at T2, samples were collected from 17 SOT-Rs and 12 HDs. The median (IQR) days between the second and third vaccine dose in SOT-Rs and HDs were 177 (175–182) and 179 (177–185) days, respectively.

Among SOT-Rs, 28% (*n* = 9) were LuT-Rs and 72% (*n* = 23) were KT-Rs. As shown in Table 1, all SOT-Rs reported at least one underlying comorbidity. As antirejection chronic immunosuppressive therapy, a combination of a calcineurin inhibitor, mycophenolate mofetil (MMF) and prednisone was reported. Moreover, 81.3% (26/32) of SOT-Rs, received low-dose steroid and calcineurin inhibitors treatments at all vaccination time points, specifically 100% of LuT-Rs and 74% of KT-Rs. Also, 68.7% of SOT-Rs receiving mycophenolate, specifically 66.6% of LuT-Rs and 74% of KT-Rs. Finally, 8 SOT-Rs were transplanted less than 2 years before vaccination (5 LuT-Rs and 3 KT-Rs, median age (IQR) 58.5 (44–63.5) years) while 24 SOT-Rs were transplanted more than 2 years before vaccination (4 LuT-Rs and 20 KT-Rs, median age (IQR) 54 (48.2–61) years).

### 3.2. Specific Humoral Response in the Study Population

At T0, all the enrolled SOT-Rs and HDs had a negative SARS-CoV-2 N-protein IgG serology test. Overall, in SOT-Rs detectable levels of anti-S antibodies were observed in 44% (14/32) at T0, in 81% (26/32) at T1 and in 88% (15/17) at T2, compared to HDs in which 100% showed detectable levels of anti-S antibody at all time points. At each time point, lower anti-S antibody titers in SOT-Rs compared to HDs were observed (T0: *p* < 0.0001, T1: *p* = 0.0495 and T2: *p* = 0.0320) (Figure 2A, Table 2).

When stratifying SOT-Rs according to the transplanted organ, lower anti-S antibody titers in LuT-Rs compared to KT-Rs at all time points were observed (T0: *p* < 0.0001, T1: *p* = 0.0008 and T2: *p* = 0.0111) (Figure 2B, Table 3). Similarly, in LuT-Rs lower anti-S antibody titers compared to HDs were observed at each time point (T0: *p* < 0.0001, T1: *p* = 0.0004 and T2: *p* = 0.0116), whereas in KT-Rs a lower anti-S antibody titer compared to HDs was observed only at T0 (*p* = 0.0232) (Figure 2B, Table 3).

Finally, in SOT-Rs being transplanted <2 years before vaccination, lower anti-S antibody titers were observed at T0 and T1 compared to SOT-Rs transplanted >2 years before vaccination (*p* = 0.0055 and *p* = 0.0159, respectively) (Figure 2C, Table 4). Conversely, no significant differences at T2 were found (Figure 2C, Table 4).

Compared to HDs, SOT-Rs transplanted < 2 years before vaccination showed lower anti-S antibody titers at T0 and T1 (*p* < 0.0001 and *p* = 0.0020, respectively). Conversely, no differences were observed at T2 in SOT-Rs transplanted < 2 years before vaccination (Figure 2C, Table 4).

SOT-Rs transplanted >2 years before vaccination showed a lower anti-S antibody titer compared to HDs only at T0 (*p* = 0.0075) (Figure 2C, Table 4), whereas no differences were observed at T1 and T2 (Figure 2C).

Overall, at T0 we observed a lower percentage of SOT-Rs (7/32) compared to HDs (9/12) having anti-S antibody titers over 264 BAU (21.9% versus 75%, respectively; *p* = 0.0034). Conversely, at both T1 and T2, no significant differences between the two groups were observed (T1: 21/32 SOT-Rs and 11/12 HDs; T2: 11/17 SOT-Rs and 12/12 HDs). Finally, at T2, a lower percentage of SOT-Rs (6/17) compared to HDs (9/12) having anti-S antibody titers over 1000 BAU was found, although statistical significance was not reached (35.3% versus 75.0%, respectively; *p* = 0.0604).

### 3.3. Specific T-Cell Response in Study Population

The evaluation of specific T-cell response showed significantly lower percentages of CD4+ and CD8+ responding T-cells in SOT-Rs compared to HDs at all time points (T0: *p* < 0.0001 and *p* = 0.0176, respectively; T1: *p* < 0.0001 and *p* = 0.0342, respectively; T2: *p* = 0.0383 and *p* = 0.0027, respectively) (Figure 3A, Table 2). Similarly, lower percentages of CD4+ and CD8+ polyfunctional T-cells in SOT-Rs compared to HDs were observed at all time points (T0: *p* < 0.0001 and *p* = 0.0024; T1: *p* = 0.0006 and *p* < 0.0001, respectively; T2: *p* = 0.0055 and *p* = 0.0087, respectively) (Figure 2A, Table 2).

No statistically significant differences in the percentages of responding and polyfunctional T-cells were observed by comparing LuT-Rs and KT-Rs at all time points (Figure 3B, Table 3).

Comparing the two SOT-R subgroups (LuT-Rs and KT-Rs) to HDs, a lower percentage of responding and polyfunctional T-cells was observed at all time points (Figure 3B, Table 3). All data and *p* values are reported in Table 3.

Finally, no statistically significant differences were found at each time point in the percentage of responding T-cells between SOT-Rs being transplanted <2 years and >2 years after vaccination (Figure 3C, Table 4). Comparing these two SOT-R subgroups to HDs, a lower percentage of responding and polyfunctional T-cells in SOT-Rs was observed at all time points. All data and *p* values are reported in Table 4.

Regarding the T-cell response quality, an uneven T-cell subset distribution in SOT-Rs compared to HDs was observed at both T0 and T1 (Figure 3D). Specifically, among SOT-Rs a predominance of IFNγ-IL2+TNFα--producing T-cells, followed by IFNγ-IL2-TNFα+-producing T-cells was observed, whereas a heterogeneous distribution of cytokine-producing T-cells was found in HDs (Figure 3D). Conversely, at T2 a heterogeneous distribution of cytokine-producing T-cells was reported in both SOT-Rs and HDs (Figure 3D).

### 3.4. Evaluation of S-Specific Omicron B.1.1.529/BA.5 T-Cell Response

An evaluation exploring differences in specific T-cell responses against SARS-CoV-2 wild-type (wt) and Omicron B.1.1.529/BA.5 (Omicron) S proteins was performed in 7 SOT-Rs and 12 HDs at T2. No statistically significant differences in the percentage of responding and polyfunctional T-cells were observed in both SOT-Rs and HDs when comparing the response to SARS-CoV-2 wt and Omicron variant (Figure 4A,B, Table 5).

Finally, comparing SOT-Rs and HDs, lower percentages of responding T-cells against both SARS-CoV-2 wild-type and Omicron variant stimulation were found (CD4: *p* = 0.0048 and *p* = 0.0070, respectively; CD8: *p* = 0.0011 and *p* = 0.0009, respectively), as well as in the percentage of polyfunctional T-cells (CD4: *p* < 0.0001 and *p* = 0.0105, respectively; CD8: *p* = 0.0141 and *p* = 0.0141, respectively) (Figure 4A,B).

## 4. Discussion

After the introduction of SARS-CoV-2 vaccines, increasing interest in vaccine immunogenicity has been reported in both immunocompetent and immunocompromised subjects [1]. However, considering the immunocompromised status of frail population groups, such as SOT-Rs, reduced responses to vaccination have been observed [3,15,28,29].

BNT162b2 vaccination has been shown to induce a lower immune response in SOT-Rs. Conversely, it remains unclear how different transplanted organs as well as the time since transplantation may affect vaccine immunogenicity after a third BNT162b2 dose, which is especially important given the emergence of the Omicron sublineages of SARS-CoV-2.

As reported by Willauer et al. [30], antibody detectable levels after two doses of mRNA vaccine were reached only in 59% of SOT-Rs compared to healthy subjects in which a 100% seroconversion rate was observed. However, after the third dose of the mRNA vaccine, an increase of over 30% seroconversion rate has been reported, underlining the importance of booster doses [30]. In line with these data, our results suggest that an expansion of the humoral response following the third dose of the mRNA vaccine is present in SOT-Rs. Indeed, our study evidences the persistence of the humoral response elicited by the mRNA vaccine, although this response is inferior to that of HDs. However, in accordance with Willauer et al. [30], SOT-Rs showed an increased seroconversion rate at 12 months since the third dose of vaccine. These findings are also confirmed by the stratification of SOT-Rs and HDs according to both cut-offs of 264 and 1000 BAU/mL proposed by other authors as vaccine efficacy against symptomatic infection with Alpha and Omicron variant of SARS-CoV-2, respectively [31,32].

To the best of our knowledge, no study has formally compared humoral and T-cell-mediated response to anti-SARS-CoV-2 vaccine between KT-Rs and LuT-Rs, but comparisons across separate studies indicate that the rates of response are higher in KT-Rs [24]. Our observed differences between LuT-Rs and KT-Rs in humoral response could be due to the different induction therapies and the time since transplantation. These assumptions are in line with other studies on influenza vaccine showing a lower antibody response in Lu-TRs between 13 and 60 months post-transplant compared to those before transplant [33].

Besides humoral response, T-cell-mediated response in immunocompromised subjects provides a protective role in controlling SARS-CoV-2 infection by reducing clinical pathogenicity and related morbidity [28,34]. As reported by Vafea et al., in SOT-Rs the T-cell responses might be elicited even in the absence of a humoral response to the SARS-CoV-2 vaccine, suggesting that SOT-Rs might potentially remain protected against COVID-19 [35].

Overall, we observed lower percentages of both responding and polyfunctional T-cells in SOT-Rs compared to HDs at all time points. Similarly, lower responses were observed also after stratifying SOT-Rs according to the type of organ transplanted and the time elapsed since the transplant.

Given that, we performed a broad characterization of the functional profiles of specific CD4+ and CD8+ T-cell responses before and after the third dose of the SARS-CoV-2 mRNA-based vaccine. In our setting, an uneven T-cell subset distribution was observed, in which IFNγ-IL2-TNFα+ and IFNγ-IL2+TNFα--producing T-cells prevailed. Since the quality of T-cell response, characterized by a multifunctional profile, has been associated with higher protection in response to infections and vaccinations [6,7,8,36,37], the highly uneven subset distribution of T-cells found in our cohort might indicate a qualitatively inferior T-cell response to vaccination. However, after twelve months since the third dose of the SARS-CoV-2 mRNA-based vaccine, a dynamic change in the subset distribution was observed, in which a more heterogeneous distribution, like HD ones, was observed in SOT-Rs as well. The exact reason for this phenomenon is unknown to us; however, it is in line with data reported by other authors showing an increased rate of IFN-γ release by T cells at 12-month follow-up among SOT-Rs, the least produced cytokine at earlier times in our study [38].

In addition, we performed an evaluation of the mRNA-BNT162b2 SARS-CoV-2 vaccine immune response against Omicron B.1.1.529/BA.5 strains. In line with other studies evaluating immune response to the Omicron variant in SOT-Rs [22,29,39], in our study the percentage of CD4+ and CD8+ responding and polyfunctional T-cells did not differ comparing wild-type and Omicron variants, suggesting a preservation of the T-cell response against this variant. Indeed, as reported by Willauer et al. and by Moss et al. [30,40], although SARS-CoV-2 variants could partially evade humoral immunity, the T-cell-mediated immunity appears to be retained.

Our study has several limitations, including the low number of SOT-Rs and the uneven sample sizes when stratifying the SOT-R population for the transplanted organ and for the time elapsed since the transplant. Moreover, we excluded asymptomatic SARS-CoV-2 infections using lack of IgG responses against the N-protein. The natural course of SARS-CoV-2 infection involves the appearance and persistence of anti-N and anti-S antibodies for several months post-infection [41,42,43,44]. For instance, the anti-N antibody has a shorter half-life and persistence compared to anti-S antibodies [42], with the anti-N antibody peaking at around 30 days post-infection [41]. However, with the global vaccination campaign and the emergence of new variants, it is increasingly common that SARS-CoV-2 infection occurs in already vaccinated or previously infected subjects [45,46]. This scenario further influences anti-N and anti-S antibody kinetics. Indeed, patients with previous vaccination may not reliably induce robust anti-N IgG responses, and the use of anti-N antibody responses as a surrogate for recent infection may not be reliable for surveillance of COVID-19 in the era of hybrid immunity expansion [47].

Collectively, our data support the hypothesis that in SOT-Rs the administration of SARS-CoV-2 mRNA-based vaccine might elicit a weaker humoral and cellular immune response compared to HDs, suggesting that in SOT-Rs the third vaccine dose effect might wane during a 1-year observation period and perhaps booster doses should be considered. Moreover, in our study, the organ transplanted as well as the time elapsed since transplant affects vaccine immunogenicity after a third BNT162b2 dose in SOT-Rs, especially in antibody levels. However, T-cell-mediated response seems to be retained which is mainly important given the emergence of the Omicron sublineages of SARS-CoV-2.

Furthermore, in SOT-Rs, SARS-CoV-2 breakthrough infections after vaccination have been reported [20]. In this context, post-vaccination testing for humoral and cellular-mediated immune responses might be needed to determine a more personalized preventive strategy against SARS-CoV-2, such as vaccine booster doses or the use of prophylactic monoclonal antibodies. Indeed, recent data suggest that monoclonal antibody prophylactic therapy elicits a strong humoral response against SARS-CoV-2 in SOT-Rs [48,49,50]. Finally, the use of mRNA booster vaccines has been shown to elicit a stronger and broader immune response to SARS-CoV-2 Omicron variants compared to the wild-type monovalent booster vaccines [51,52]. Therefore, we conclude that the ongoing longitudinal evaluation of SARS-CoV-2 humoral and cellular responses to mRNA-based vaccine in SOT-Rs might be valuable and necessary to allow the frequent re-evaluation of an ever-changing viral landscape in immunocompromised individuals.

## 5. Conclusions

In view of the emergence of Omicron sublines of SARS-CoV-2, in SOT-Rs the administration of an additional dose of mRNA-based SARS-CoV-2 vaccine should be considered to enhance the humoral and cellular immune response. In addition, the transplanted organ (probably due to the different induction therapy) and the time since transplantation influence the immunogenicity of the vaccine, especially the antibody levels.

## Figures and Tables

**Figure 1 vaccines-12-00224-f001:**
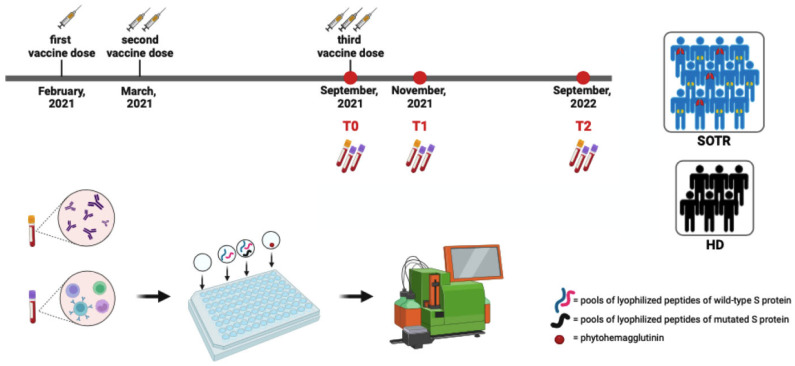
Timeline and schematic representation of the study design. T0: before third mRNA vaccine dose, T1: two months after the third mRNA vaccine dose, T2: twelve months after the third mRNA vaccine dose, SOT-R: solid organ transplant recipient, HD: healthy donor, wild-type, S: Wuhan, mutated, S: Omicron variant.

**Figure 2 vaccines-12-00224-f002:**
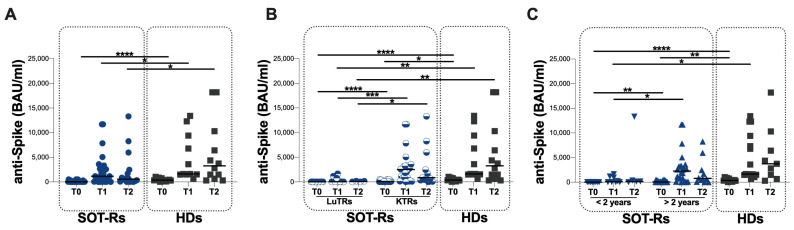
Anti-S antibody titers in SOTR and HD (**A**) and in SOTR stratified according to the type of organ transplanted (**B**) and the time elapsed since transplant (**C**). SOT-Rs: solid organ transplant recipients, HDs: healthy donors, BAU: binding antibody unit, LuT-Rs: lung transplant recipients, KT-Rs: kidney transplant recipients, T0: before third mRNA vaccine dose, T1: two months after the third mRNA vaccine dose, T2: twelve months after the third mRNA vaccine dose, <2 years: transplantation less 2 years before vaccination, >2 years: transplantation more than 2 years before vaccination. *: 0.05 < *p* < 0.01; **: 0.01 < *p* < 0.001; ***: 0.001 < *p* < 0.0001; ****: *p* > 0.0001.

**Figure 3 vaccines-12-00224-f003:**
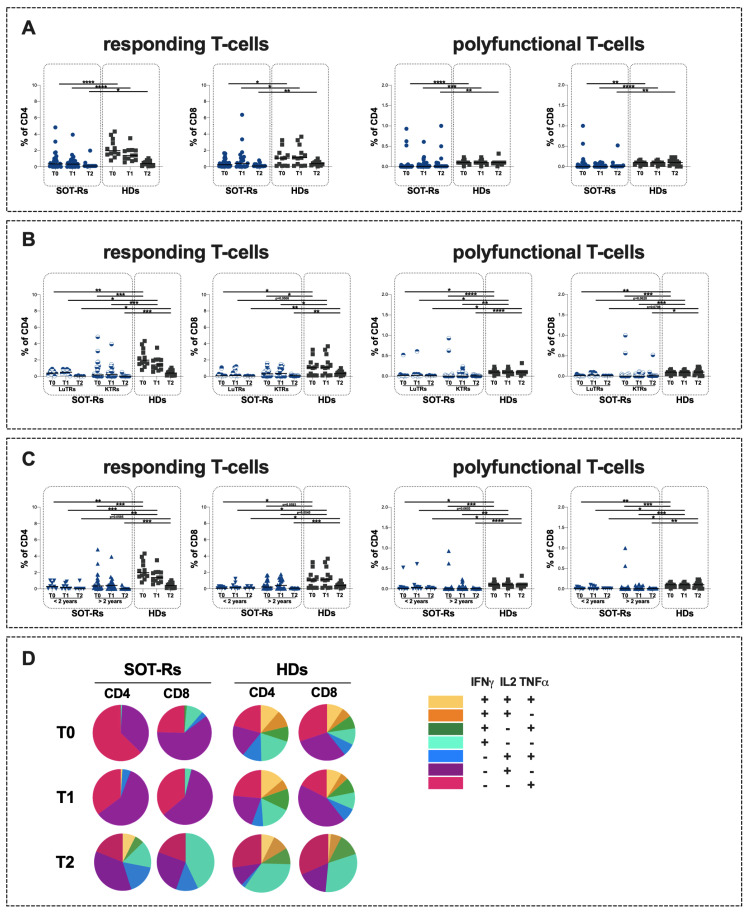
Responding and polyfunctional T-cell response in SOT-Rs and HDs (**A**) and in SOT-Rs stratified according to the type of organ transplanted (**B**) and the time elapsed since transplant (**C**) and T-cell response quality in SOT-Rs and HDs (**D**). SOT-Rs: solid organ transplant recipients, HDs: healthy donors, LuT-Rs: lung transplant recipients, KT-Rs: kidney transplant recipients, T0: before third mRNA vaccine dose, T1: two months after the third mRNA vaccine dose, T2: twelve months after the third mRNA vaccine dose, <2 years: solid organ transplant recipient receiving transplant less 2 years before vaccination; >2 years: solid organ transplant recipient receiving transplant more than 2 years before vaccination. *: 0.05 < *p* < 0.01; **: 0.01 < *p* < 0.001; ***: 0.001 < *p* < 0.0001; ****: *p* > 0.0001.

**Figure 4 vaccines-12-00224-f004:**
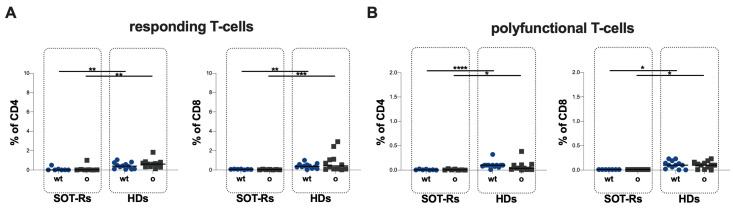
Responding (**A**) and polyfunctional T-cell response (**B**) to wild-type and Omicron stimulation at T2. SOT-Rs: solid organ transplant recipients, HDs: healthy donors, wt: wild-type, o: Omicron variant. *: 0.05 < *p* < 0.01; **: 0.01 < *p* < 0.001; ***: 0.001 < *p* < 0.0001; ****: *p* > 0.0001.

**Table 1 vaccines-12-00224-t001:** Clinical and demographic characteristics of the study population.

	SOT-Rs (*n* = 32)	LuT-Rs (*n* = 9)	KT-Rs (*n* = 23)	HDs (*n* = 12)
**Age, years**	56 (48–61)	56 (46–62)	55 (49–61)	50 (47–58)
**Male/female**	22/10	7/2	15/8	7/5
**Time elapsed since transplant, months**	76 (42–162)	20 (13–96)	76 (60–181)	
**Immunosuppressive treatment**				
Steroids	26	9	17	
Antimetabolites	22	6	16	
Calcineurin inhibitors	32	9	23	
mTOR inhibitors	1	0	1	
**Laboratory data**				
WBC (×10^9^/L)	7.2 (6.1–9.4)	8.2 (5.4–10.0)	7.2 (6.2–9.5)	
Neutrophils (×10^9^/L)	4.7 (3.4–6.4)	5.1 (3.7–6.7)	4.5 (3.4–6.4)	
Lymphocytes (×10^9^/L)	2.2 (1.4–2.4)	1.4 (1.3–2.6)	2.2 (1.7–2.5)	
PLT (×10^9^/L)	212 (156–244)	243 (145–265)	210 (159–240)	
Creatinine (mg/dL)	1.6 (1.3–2.0)	1.3 (1.3–2.1)	1.7 (1.3–1.9)	
Azotemia (mg/dL)	55 (40–79)	54 (8–85)	58 (42–78)	
**Comorbidities**				
Diabetes	7	4	3	
Arterial hypertension	26	5	21	
Dyslipidemia	15	1	14	
Cardiopathy	4	2	2	

Data are shown as median (interquartile range, IQR). *n*: number; SOT-Rs: solid organ transplant recipients; HDs: healthy donors; LuT-Rs: lung transplant recipient; KT-Rs: kidney transplant recipient; WBC: whole blood cells; PLT: platelets.

**Table 2 vaccines-12-00224-t002:** Immunological data in the study population.

	SOT-Rs (*n* = 32)			HDs (*n* = 12)		
	T0	T1	T2	T0	T1	T2
**anti-S antibody titers (BAU/mL)**	31 (23–149)	1150 (101–3050)	562 (80–2270)	369 (189–701)	1610 (1520–9400)	3270 (869–9328)
**responding T-cells**						
% of CD4+	0.36 (0.09–1.01)	0.34 (0.00–0.72)	0.10 (0.10–0.10)	1.98 (1.52–3.29)	1.39 (0.88–2.04)	0.41 (0.11–0.72)
% of CD8+	0.25 (0.05–0.64)	0.41 (0.04–1.10)	0.09 (0.06–0.20)	1.06 (0.11–1.63)	1.12 (0.18–2.46)	0.38 (0.20–0.57)
**polyfunctional T-cells**						
% of CD4+	0.00 (0.00–0.02)	0.01 (0.00–0.05)	0.01 (0.01–0.09)	0.10 (0.10–0.13)	0.10 (0.10–0.13)	0.10 (0.08–0.10)
% of CD8+	0.00 (0.00–0.05)	0.00 (0.00–0.06)	0.01 (0.01–0.01)	0.10 (0.05–0.11)	0.10 (0.05–0.11)	0.10 (0.03–0.18)

Data are shown as median (interquartile range, IQR). *n*: number; BAU: binding antibody unit; SOT-Rs: solid organ transplant recipients; HDs: healthy donors; T0: before third mRNA vaccine dose; T1: two months after the third mRNA vaccine dose; T2: twelve months after the third mRNA vaccine dose.

**Table 3 vaccines-12-00224-t003:** Immunological data in SOT-Rs stratified according to the type of organ transplanted.

	LuT-Rs (*n* = 9)			KT-Rs (*n* = 23)			HDs (*n* = 12)			*p* Value *	
	T0	T1	T2	T0	T1	T2	T0	T1	T2	LuT-Rs vs. HD	KT-Rs vs. HD
**anti-S antibody titers (BAU/mL)**	4.8 (4.8–16.2)	23 (7–956)	70.3 (7–144)	88.4 (31–289)	2500 (862–3320)	870 (399–7110)	369 (189–701)	1610 (1520–9400)	3270 (869–9328)	T0: <0.0001 T1: 0.0004 T2: 0.0116	T0: 0.0232 T1: ns T2: ns
**responding T-cells**										T0: 0.0025 T1: 0.0322 T2: 0.0360	T0: 0.0003 T1: 0.0001 T2: 0.0002
% of CD4+	0.39 (0.20–0.69)	0.50 (0.32–0.81)	0.02 (0.01–0.08)	0.17 (0.07–1.13)	0.21 (0.0–0.65)	0.01 (0.01–0.05)	1.98 (1.52–3.29)	1.39 (0.88–2.04)	0.41 (0.11–0.72)		
% of CD8+	0.07 (0.05–0.63)	0.09 (0.05–0.93)	0.03 (0.01–0.06)	0.30 (0.02–0.65)	0.35 (0.0–1.0)	0.08 (0.06–0.10)	1.06 (0.11–1.63)	1.12 (0.18–2.46)	0.38 (0.20–0.57)	T0: 0.0371 T1:0.0508 T2: 0.0011	T0: 0.0492 T1: 0.0306 T2: 0.0021
**polyfunctional T-cells**										T0: 0.0498 T1: 0.0415 T2: 0.0286	T0: <0.0001 T1: 0.0025 T2: <0.0001
% of CD4+	0.02 (0.0–0.03)	0.02 (0.0–0.05)	0.01 (0.01–0.02)	0.0 (0.0–0.01)	0.0 (0.0–0.08)	0.01 (0.0–0.01)	0.10 (0.10–0.13)	0.10 (0.10–0.13)	0.10 (0.08–0.10)		
% of CD8+	0.01 (0.0–0.02)	0.03 (0.0–0.09)	0.01 (0.01–0.01)	0.0 (0.0–0.03)	0.0 (0.0–0.04)	0.01 (0.01–0.02)	0.10 (0.05–0.11)	0.10 (0.05–0.11)	0.10 (0.03–0.18)	T0: 0.0052 T1: 0.0628 T2: 0.0798	T0: 0.0003 T1: 0.0002 T2: 0.0470

Data are shown as median (interquartile range, IQR). *n*: number; BAU: binding antibody unit; HDs: healthy donors; LuT-Rs: lung transplant recipients; KT-Rs: kidney transplant recipients; T0: before third mRNA vaccine dose; T1: two months after the third mRNA vaccine dose; T2: twelve months after the third mRNA vaccine dose; ns: not significant. *: The nonparametric comparative Kruskal–Wallis test was used for comparing medians of LuT-Rs and KT-Rs with HDs at each time point. Dunn’s multiple comparison post-test was used for comparing medians of LuT-Rs with HDs as well as KT-Rs with HDs.

**Table 4 vaccines-12-00224-t004:** Immunological data in SOT-Rs stratified according to the time since transplant.

	<2 Years since Tx (*n* = 8)			>2 Years since Tx (*n* = 24)			HDs (*n* = 12)			*p* Value *	
	T0	T1	T2	T0	T1	T2	T0	T1	T2	<2 Years since Tx vs. HD	>2 Years since Tx vs. HDs
**anti-S antibody titers (BAU/mL)**	16 (4.8–31)	122 (11–1093)	149 (70–6820)	62 (31–287)	2270 (286–3300)	771 (163–5085)	369 (189–701)	1610 (1520–9400)	3270 (869–9328)	T0: <0.0001 T1: 0.0020 T2: ns	T0: 0.0075 T1: ns T2: ns
**responding T-cells**										T0: 0.0020 T1: 0.0008 T2: 0.0545	T0: 0.0004 T1: 0.0013 T2: 0.0007
% of CD4+	0.30 (0.06–0.94)	0.17 (0.0–0.63)	0.01 (0.01–0.51)	0.35 (0.10–1.10)	042 (0.08–0.75)	0.01 (0.01–0.07)	1.98 (1.52–3.29)	1.39 (0.88–2.04)	0.41 (0.11–0.72)		
% of CD8+	0.06 (0.06–0.27)	0.12 (0.05–0.67)	0.08 (0.02–0.31)	0.20 (0.02–0.98)	0.40 (0.02–1.09)	0.07 (0.06–0.09)	1.06 (0.11–1.63)	1.12 (0.18–2.46)	0.38 (0.20–0.57)	T0: 0.0137 T1: 0.0416 T2: 0.0385	T0: 0.0593 T1: 0.0545 T2: 0.0006
**polyfunctional T-cells**										T0: 0.0212 T1: 0.0603 T2: 0.0211	T0: 0.0001 T1: 0.0022 T2: <0.0001
% of CD4+	0.01 (0.0–0.04)	0.03 (0.0–0.08)	0.01 (0.01–0.03)	0.0 (0.0–0.01)	0.0 (0.0–0.05)	0.01 (0.0–0.01)	0.10 (0.10–0.13)	0.10 (0.10–0.13)	0.10 (0.08–0.10)		
% of CD8+	0.01 (0.0–0.04)	0.0 (0.0–0.07)	0.01 (0.01–0.01)	0.0 (0.0–0.02)	0.0 (0.0–0.06)	0.01 (0.01–0.01)	0.10 (0.05–0.11)	0.10 (0.05–0.11)	0.10 (0.03–0.18)	T0: 0.0059 T1: 0.0120 T2: 0.0236	T0: 0.0003 T1: 0.0006 T2: 0.0080

Data are shown as median (interquartile range, IQR). *n*: number; BAU: binding antibody unit; <2 years since Tx: transplantation less 2 years before vaccination, >2 years since Tx: transplantation more than 2 years before vaccination; HD: healthy donor; Tx: organ transplant; T0: before third mRNA vaccine dose; T1: two months after the third mRNA vaccine dose; T2: twelve months after the third mRNA vaccine dose; ns: not significant. *: The nonparametric comparative Kruskal–Wallis test was used for comparing medians of <2 years since Tx and >2 years since Tx groups with HDs at each time point. Dunn’s multiple comparison post-test was used for comparing medians of <2 years since Tx group with HDs as well as >2 years since Tx group with HDs.

**Table 5 vaccines-12-00224-t005:** Specific T-cell responses against wild-type and Omicron B.1.1.529/BA.5 S proteins in SOT-Rs and HDs.

	SOT-Rs (*n* = 7)		HDs (*n* = 12)	
	wt	o	wt	o
**responding T-cells**				
% of CD4+	0.01 (0.01–0.10)	0.02 (0.01–0.06)	0.41 (0.11–0.72)	0.61 (0.33–0.78)
% of CD8+	0.08 (0.05–0.10)	0.04 (0.02–0.06)	0.38 (0.20–0.57)	0.43 (0.10–1.10)
**polyfunctional T-cells**				
% of CD4+	0.01 (0.0–0.02)	0.01 (0.0–0.02)	0.10 (0.08–0.10)	0.04 (0.01–0.11)
% of CD8+	0.01 (0.01–0.01)	0.01 (0.01–0.01)	0.10 (0.03–0.18)	0.10 (0.02–0.16)

Data are shown as median (interquartile range, IQR). *n*: number; BAU: binding antibody unit; SOT-Rs: solid organ transplant recipients; HDs: healthy donors; wt: wild-type, o: Omicron variant.

## Data Availability

The raw data supporting the conclusions of this article will be made available by the authors without undue reservation.

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
