# Peer review of "The Long-Term Immunogenicity of mRNABNT162b Third Vaccine Dose in Solid Organ Transplant Recipients"

_vaccines, 2024, doi:10.3390/vaccines12030224_

Round 1

Reviewer 1 Report

Comments and Suggestions for Authors

This manuscript describes the long-term immunogenicity of an mRNA-based vaccine against SARS-CoV-2 in immunosuppressed solid-organ transplant (SOT) recipients and compares the results to those seen in healthy individuals. The authors address a valid question, as the level of the long-term immune responses in SOT-recipients will likely correlate with the level of protection against severe disease manifestations in this group, compared to healthy individuals. There are, however, some issues with the manuscript detailed below.

Specific comments

Major

1.       The title of the manuscript is misleading, as the work addresses the long-term immunogenicity of the vaccine rather than the efficacy.

2.       It is well established that the lack of IgG responses against the N-protein of SARS-CoV-2 are unreliable as a criterion to exclude prior infection. Thus the authors cannot exclude pre-exposure to the virus, based on this assay. There is also no information on a past medical history of the participants with regard to COVID-19.

3.       Results in lines 210-232 are described in a confusing text format and could have been presented in a much clearer fashion in a table format.

4.       It is not immediately clear, what the novel information, provided by this work is, compared to published work of others. This should have been more clearly explained.

Minor

-          The term “healthy donor” is not defined (blood donors?)

-          Line 277, “underlying” should likely read “underlining”.

Comments on the Quality of English Language

Minor amendments required

Reviewer 2 Report

Comments and Suggestions for Authors

An interesting study of humoral and cellular immune response to 3 doses vaccination with m-RNA vaccine in solid organ transplant as compared to matched healthy control with up to twelve months. I suggest to the authors to quantify (percentage)the number of patients and controls at each time having anti-spike antibodies > 264 BAU (a correlate of protection against Wuhan strain and first variants) and > 1000 BAU (proposed for Omicron)

Comments on the Quality of English Language

English language should be checked by a native translator to improve the quality of the text

Reviewer 3 Report

Comments and Suggestions for Authors

  In the current study, the authors investigated humoral and T-cell responses in solid organ transplant recipients (SOT-Rs) in comparison with healthy donors (HDs) at three time-points after the third dose administration of SARS-CoV-2 mRNA vaccine. Overall, humoral and T-cell responses in SOT-Rs were weaker than those in HDs, suggesting that the third vaccine dose effect in SOT-Rs might wane during a 1-year observation period and booster doses should be considered. This manuscript is well-written and contains useful information to perform COVID-19 vaccine strategies for immunocompromised individuals. I have no serious criticisms in the current version. However, I have raised two points which need to be clarified. There are given below.

Specific points:

1)     It was described that lower anti-S antibody titers in LuTRs compared to KTRs at all time-points were observed (lines 167-168). Please explain the possible reasons for such a difference in the Discussion.

2)     A highly uneven T-cell subset distribution in SOT-Rs was observed at both T0 and T1, indicating a qualitatively inferior T-cells, whereas a heterogenous distribution of cytokine producing T-cell was found in HDs (lines 233-236). Please describe the possible reasons why such a phenomenon occurred. 

Round 2

Reviewer 1 Report

Comments and Suggestions for Authors

Accept current version

Author Response

As suggested, we have reworded the "Material and method" part as much as possible